# RecDCL: Dual Contrastive Learning for Recommendation

## ABSTRACT

Self-supervised recommendation (SSR) has achieved great success in mining the potential interacted behaviors for collaborative filtering in recent years. As a major branch, Contrastive Learning (CL) based SSR conquers data sparsity in Web platforms by contrasting the embedding between raw data and augmented data. However, existing CL-based SSR methods mostly focus on contrasting in a batch-wise way, failing to exploit potential regularity in the feature-wise dimension, leading to redundant solutions during the representation learning process of users (items) from Websites. Furthermore, the joint benefits of utilizing both Batch-wise CL (BCL) and Feature-wise CL (FCL) for recommendations remain underexplored. To address these issues, we investigate the relationship of objectives between BCL and FCL. Our study suggests a cooperative benefit of employing both methods, as evidenced from theoretical and experimental perspectives. Based on these insights, we propose a dual CL method for recommendation, referred to as RecDCL. RecDCL first eliminates redundant solutions on user-item positive pairs in a feature-wise manner. It then optimizes the uniform distributions within users and items using a polynomial kernel from an FCL perspective. Finally, it generates contrastive embedding on output vectors in a batch-wise objective. We conduct experiments on four widely-used benchmarks and an industrial dataset. The results consistently demonstrate that the proposed RecDCL outperforms the state-of-the-art GNNs-based and SSL-based models (with up to a 5.65% improvement in terms of Recall@20), thereby confirming the effectiveness of the joint-wise objective. All source codes used in this paper are publicly available[1].

## KEYWORDS

Recommender Systems, Self-supervised Learning, Feature-wise Contrastive Learning, Batch-wise Contrastive Learning

**ACM Reference Format:**

Anonymous Author(s). 2023. RecDCL: Dual Contrastive Learning for Recommendation. In *Submission to WWW '24: TheWebConf, May 13–17, 2024, Singapore*. ACM, New York, NY, USA, 17 pages. https://doi.org/XXXXXXX.XXXXXXX

**Relevance to the Web and to the track.** Recommendation systems are prevalent in online Web platforms (e.g., e-commerce recommendation, music or video recommendation), aiming to mine the potential interactions. This work focuses on exploring embedding information and eliminating redundant solutions for users and items in online platforms- - -the representation methods of the Web data that have been previously caused- - -by designing RecDCL, a dual contrastive learning method to facilitate the development of Web recommendation systems with experiments on Beauty, Food, Game, Yelp, and online payment data. RecDCL directly corresponds to the topics of "recommender systems" in the CFP of *User Modeling and Recommendation* track.

## 1 INTRODUCTION

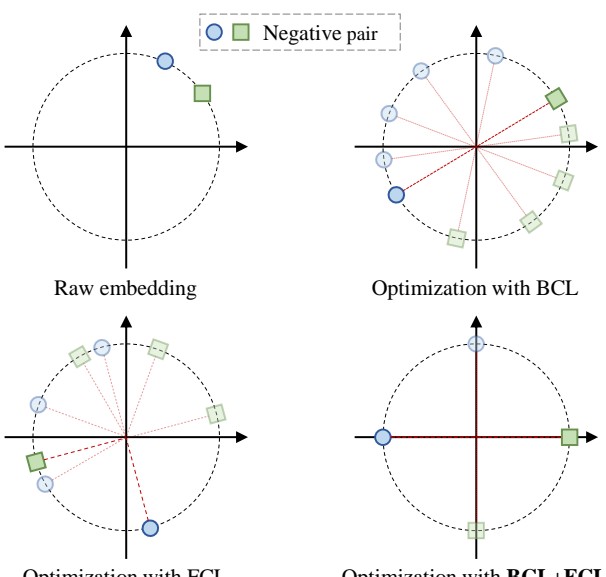

**Figure 1: The motivating example that shows the effect for a negative pair in BCL, FCL, and BCL+FCL, where the light-shaded symbols indicate potentially possible solutions.** *In this example, BCL (top right) tends to align the negative pair on a straight line, i.e., even distribution in a circle; FCL (bottom left) mostly encourages the two representations to be orthogonal; BCL+FCL (bottom right) drives the two samples to saturate on either the x axis or the y axis. Note that using either BCL alone or FCL alone will result in infinite potential solutions. In contrast, combining BCL and FCL yields only four possible solutions: $\{(0,1),(0,-1)\}$, $\{(0,-1),(0,1)\}$, $\{(1,0),(-1,0)\}$ and $\{(-1,0),(1,0)\}$, which cancels the redundant solutions but never misses an optimal solution, and thus is intuitively a more reasonable regularization compared with BCL alone or FCL alone. High-dimensional cases are analogous.*

Contrastive learning (CL) [18] is known as a major branch of self-supervised learning. The fundamental idea behind CL is to artificially augment more supervised instances and conduct a pretext task with augmented data, addressing the issue of data sparsity. In recent years, CL-based collaborative filtering (CF) has been proposed and achieved great success to mine interacted records from Web systems for recommendation [22, 33, 37, 42, 43, 49].

---

[1]https://anonymous.4open.science/r/RecDCL-FC33/

*Submission to WWW '24, May 13–17, 2024, Singapore*
© 2023 Association for Computing Machinery.
ACM ISBN 978-1-4503-XXXX-X/18/06...$XX.00
https://doi.org/XXXXXXX.XXXXXXX

                                                                                          

In general, CL-based collaborative filtering methods focus on batch-wise objective functions. Batch-wise objectives aim to maximize the similarity of embedding between positive pairs (diagonal) while minimizing that of negative pairs (off-diagonal). A typical kind of batch-wise based CF methods apply BPR loss [28] to predict the users' preferences on multiple interacted Web platforms (e.g., e-commerce, music, video), such as graph neural networks (GNNs)-based models (e.g., PinSAGE [40], LightGCN [14], and ApeGNN [46]) and self-supervised learning (SSL)-based models (e.g., SGL [37] and SimGCL [42]). In particular, these methods usually require both the user-item interacted pairs and the negative counterparts generated by negative sampling. However, since the negative sampling scheme may mistakenly treat "positive but unobserved" pairs as negative pairs, there exist critical limitations in the performance of these methods. Moreover, some recent batch-wise CL (BCL) recommendation methods (e.g., BUIR [22], CLRec [49], and DirectAU [33]) point out that more robust improvements can be obtained without negative sampling for Web recommendation systems. **However, these BCL methods may lead to trivial constant solutions since they fail to leverage embedding information of users and items from Web platforms, which is illustrated in Figure 1.**

**Table 1: Critical comparison between BCL methods, FCL methods, and RecDCL.**

| Models | Embedding Information | Sample Information | Redundant Solution | Distribution |
|--------|:---------------------:|:------------------:|:------------------:|:------------:|
| BCL | ✗ | ✓ | ✓ | Even |
| FCL | ✓ | ✗ | ✗ | Orthogonal |
| **RecDCL** | ✓ | ✓ | ✗ | Even |

To address this issue, we investigate CL-based methods in various domains and conclude their critical differences in Table 1. The objective functions of CL usually fall into two categories: batch-wise objectives and feature-wise objectives. As for the feature-wise objectives, existing works have attracted full attention in the computer vision (CV) domain. In particular, feature-wise CL methods such as Barlow Twins [44] and VICREG [1] have been devoted to investigating the importance of embedding vectors and proposing novel feature-wise objective functions. These methods maximize the variability of the embeddings by decorrelating the components in the feature-wise dimension, which can avoid collapse [35] and yield the desired performance. **However, as shown in Figure 1, these FCL methods ignore important information provided in batch-wise objectives and lead to orthogonal distribution.** In light of this, a meaningful question then arises: *Is there an effective optimization objective between batch-wise CL and feature-wise CL for self-supervised recommendations?* Regarding this, CL4CTR [34] proposes feature alignment and field uniformity and masks feature and dimension information to address the "long tail" distribution of feature frequencies for CTR prediction. **However, previous studies [34, 35] only explored the connection between BCL and FCL, there has been a lack of a native interpretation to connect them and little effort has been made to understand the effect of combining them.**

To answer the above question, we investigate a native connection of the objective between the batch-wise CL and the feature-wise CL (Figure 1 and Observation 3.1) and present a perspective to show a cooperative benefit by using them both (Observation 3.2) from the perspective of theory and experiment. Based on these analyses, we propose a dual CL method, referred to as RecDCL. RecDCL joints feature-wise objectives and batch-wise objectives for self-supervised recommendations. On one hand, RecDCL optimizes the feature-wise CL objective (FCL) by eliminating redundancy between users and items. Especially, FCL captures the distribution of user-item positive pairs by measuring a cross-correlation matrix and optimizes user (items) distribution via polynomial kernel. On the other hand, as a batch-wise dimension, we design basic BCL and advanced BCL to enhance the robustness of the representations, which the latter combines historical embedding with current embeddings and generates contrastive views via online and target networks. Extensive experiments validate that RecDCL outperforms the state-of-the-art GNN-based and SSL-based models (by up to 5.34% on Beauty), showing the effectiveness of jointly optimizing the feature-wise and batch-wise objectives.

The main contributions of this work are as follows:

- We theoretically reveal a native connection of the objective between feature-wise CL and batch-wise CL, and demonstrate a cooperative benefit by using them both from theoretical and experimental perspectives.

- Based on the above analysis, we propose a dual CL method named RecDCL with joint training objectives in feature-wise and batch-wise ways to learn informative representations.

- We conduct extensive experiments on four public datasets and one industrial dataset to show the effectiveness of the proposed RecDCL. Compared to multiple state-of-the-art methods, RecDCL achieves significant performance improvements up to 5.34% in terms of NDCG@20 on Beauty dataset and 5.65% in terms of Recall@20 on the Yelp dataset.

## 2 PRELIMINARIES

This section briefly introduces three closely related technologies: batch-wise collaborative filtering, batch-wise contrastive learning, and feature-wise contrastive learning.

## 2.1 Batch-wise Collaborative Filtering

Let $\mathcal{U}$, $\mathcal{I}$ and $\mathcal{R}$ of a user-item bipartite graph $\mathcal{G}$ be the user set, item set and interactions between users and items. A collaborative filtering method aims to rank all items and predict the possible items that the user will interact with next. A popular architecture for collaborative filtering is GCN. Given the initial embedding $\mathbf{e}_u$ and $\mathbf{e}_i$ for user $u$ and item $i$, GCN aims to iteratively perform neighborhood aggregation and update node representation on $\mathcal{G}$. In each iteration, a user node $u$ aggregates the neighbors' representation of the $(l-1)$-th layer, with which $\mathbf{e}_u^{(l-1)}$ is updated into $\mathbf{e}_u^{(l)}$. The same update rule applies to an item node $i$. After the $L$-layer iteration, the final ranking score between $u$ and $i$ is calculated via the inner product of their representations, i.e., $\hat{y}_{u,i} = \mathbf{e}_u^{(L)} \mathbf{e}_i^{(L)}$. The optimization objective is to push $\hat{y}_{u,i}$ closer to the ground truth $y_{ui}$ in a point-wise manner. As a representative implementation, pairwise Bayesian Personalized Ranking (BPR) [28] enforces the predicted score of positive interaction is higher than its negative

counterpart as follows:

$$\mathcal{L} = \sum_{(u,i,j) \in O} -\log \sigma(\hat{y}_{u,i} - \hat{y}_{u,j}), \tag{1}$$

where $O$ is the training data, $(u,i)$ is the observed interaction, $(u,j)$ is the unobserved interaction, and $\sigma$ is sigmoid function.

## 2.2 Batch-wise Contrastive Learning (BCL)

Due to the marginal improvements in existing BPR loss, a self-supervised learning paradigm has achieved great success in computer vision and natural language understanding [4, 7, 12]. Some recent studies often focus on contrastive learning dominant in the SSL-based paradigm, typically InfoNCE Loss in SGL [37], which mainly relies on the idea of applying graph augmentations on graph structure. Suppose that there are two augmented views of nodes, the views of the same node are regarded as the positive pairs, i.e., $z_u', z_u''$, and the views of any different nodes as the negative pairs, i.e., $z_u', z_v''(u, v \in \mathcal{U}, u \neq v)$. InfoNCE Loss that aims to maximize the consistency of different views of the same node and minimize that of the views among different nodes are as follows:

$$\mathcal{L} = \sum_{u \in \mathcal{U}} -\log \frac{\exp(s(z_u', z_u'')/\tau)}{\sum_{v \in \mathcal{U}} \exp(s(z_u', z_v'')/\tau)}, \tag{2}$$

in which $s(\cdot)$ means the similarity between two representation vectors. Afterwards, Cosine Contrastive Loss in BUIR [22] and SelfCF [50], Alignment and Uniformity Loss in DirectAU [33] bring more significant recommendation performance improvements. Among them, it should be particularly pointed out that the nowadays CL-based models (e.g., BUIR and DirectAU) perform top-$K$ recommendation tasks without negative sampling, which enforces faster convergence and better recommendation performance.

## 2.3 Feature-wise Contrastive Learning (FCL)

Recent studies (e.g., Barlow Twins [44] and VICREG [1]) in contrastive representation learning emphasize the importance of representations in feature-wise objectives. Given two batches of perturbed views, Barlow Twins first feeds these inputs into an encoder and produces batches of representations $\mathbf{Z}$ and $\hat{\mathbf{Z}}$, respectively. Then, it optimizes an innovative loss function by building a cross-correlation matrix $\mathbf{C}$ between $\mathbf{Z}$ and $\hat{\mathbf{Z}}$ as follows

$$\mathcal{L} = \sum_m (1 - \mathbf{C}_{mm})^2 + \lambda \sum_m \sum_{n \neq m} (\mathbf{C}_{mn})^2. \tag{3}$$

The first term, also called the invariance term, aims to make the diagonal values be close to 1 and keep the embedding invariant even if applying distortions. On the other hand, the redundancy reduction term is defined to make the off-diagonal elements be closed 0 and decorrelate each dimension representation of the embedding. In our work, we will explore contrastive representation learning via these two items for collaborative filtering.

## 3 UNDERSTANDING BCL AND FCL

Since BCL and FCL serve as the cornerstones of this paper, a more in-depth discussion about them may be warranted before diving into methodological details. To this end, we first review existing perspectives and then present our interpretation. We reveal a native connection between the two CL principles (Observation 3.1) and

discover that combining BCL and FCL intuitively forms a better regularization which can benefit from high embedding dimensions (Observation 3.2).

## 3.1 Existing Perspectives

Many BCL objectives follow a basic form known as InfoNCE [26]. The core idea of InfoNCE is to estimate model parameters by contrasting the embedding between raw samples and some randomly augmented samples, which can be approximately viewed to maximize the mutual information estimation between the two input signals. Towards a more intuitive interpretation, Wang et al. [35] show that BCL actually optimizes two quantifiable metrics of representation quality: *alignment* of features from positive pairs, and *uniformity* of the induced distribution of the (normalized) features on the hypersphere. The alignment and uniformity principle provides a nice perspective to understand BCL, which has motivated many follow-up works [6, 9, 27, 34, 38].

To achieve the desired performance, BCL usually requires some careful implementation designs such as large batch sizes [4] and memory banks [12]. To avoid these implementation details, Zbontar et al. [44] develops the first FCL method named Barlow Twins (BT), which trains the model via minimizing the redundancy between the components of representation instead of directly optimizing the geometry of embedding distribution. Kalantidis et al. [19] indicate that BT can be seen as a more modern, simpler way to optimize deep canonical correlation analysis. Tsai et al. [32] relate Barlow Twins to the Hilbert-Schmidt independence criterion, which suggests a possibility to bridge the two major families of self-supervised learning philosophies: non-contrastive and contrastive approaches.

BCL and FCL can be regarded as two dimensions of contrastive learning. Some recent works have been devoted to revealing the underlying connection between them. Towards the generalization ability of CL, Huang et al. [17] develop a mathematical analysis framework to prove that BCL and FCL enjoy some common advantages. Zhang et al. [47] propose a negative-free contrastive learning method that combines BCL and FCL, but they fail to clarify the insight for doing so. Following the principle of maximum entropy in information theory, Liu et al. [25] propose a maximum entropy coding loss, which provides a unified perspective to understand BCL and FCL objectives.

**Summary.** The explanation for BCL alone [11, 26, 35] or FCL alone [19, 32, 44] has been well established. In addition, the connection between BCL and FCL has also been explored from the generalization view [17] and the maximum entropy coding theory [25]. However, there still lacks a native interpretation to connect them without introducing extra knowledge. Furthermore, the efforts to understand the effect of combining BCL and FCL are almost missing. We try to answer these two questions next.

## 3.2 Our Interpretation

This part aims to approach two questions: what is the relationship between BCL and FCL (Cf. Observation 3.1), and why does combining them work (Cf. Observation 3.2).

Intuitively, BCL and FCL share the same mechanism, i.e., drawing positive pairs close while pushing negative pairs away. The

difference lies in the objects that make up their pairs. The considered objects are samples for BCL while features for FCL. This difference seems to endow the two CL objectives with different effects when optimizing a model. Interestingly, they actually lead the model to optimize in similar directions under some conditions. Formally, we conclude our observation as follows with theoretical analyses provided in A.1.

OBSERVATION 3.1. *If the two embedding matrices are standardized (i.e., they have mean zero and standard deviation one), then the objectives of BCL and FCL can be approximately transformed to each other*[2].

Observation 3.1 shows that there exists an inherent connection between BCL and FCL. A follow-up question would be *whether it is necessary to use them both?* We provide a perspective below to partially answer this question. The key of our perspective is to consider the role that negative pairs play in the two objectives. For BCL, it has been known that pushing the negative pairs away actually approximately encourages the samples to be evenly distributed in the embedding space, which together with the positive pair constraint implicitly enlarges the classification margin [35]. For FCL, the reason for pushing negative pairs away in the feature-wise space may be not obvious. Although some explanations have been made from the information theory [1, 21, 25, 44], a more intuitive explanation can help to understand how this regularization contributes to final embedding. Therefore, we provide an illustrative example to interpret the effect of BCL, FCL, and BCL+FCL in Figure 1, with main observations concluded in Observation 3.2.

OBSERVATION 3.2. *For normalized sample embedding, pushing negative pairs away has different influences on embedding learning between BCL and FCL. For BCL, it encourages samples to be evenly distributed in the embedding space. For FCL, it tends to drive the representations of samples to be orthogonal. This difference is mainly due to that BCL encourages the inner product of negative pairs (in the batch dimension) to be as small as possible; but FCL only enforces the inner product of negative pairs (in the feature dimension) to be close to zero, which implicitly encourages the representations of samples (in the batch dimension) to be orthogonal. If we combine BCL and FCL, pushing negative pairs away will not only encourage sample representations to be evenly distributed in the embedding space but also help eliminate redundant solutions (Cf. Figure 1)*[3]. *This regularity can benefit embedding learning as the embedding dimension increases*[4].

## 3.3 Recommendation Intuition

To validate the effectiveness of regularity in Section 3.2, we conduct an ablation study to see whether combining BCL and FCL results in a more desirable embedding distribution. Specifically, we compare the average entropy of the embeddings among FCL, BCL, and BCL+FCL on the Yelp dataset. Let $\mathbf{x}$ denote the embedding of a sample. We select the top-$K$ (1024 and 2048) absolute values of $\mathbf{x}$ via two approaches and normalize them into a K-dimensional probability distribution. The first one is we sort the embedding values

in a descending way for each sample and obtain the top-$K$ values, called each-sample. The second one is we calculate the mean values in each dimension for all samples, sort the values and obtain the top-$K$ indices, and extract the top-$K$ values for each sample, called mean-sample. Based on the above step, the entropy of each sample can be calculated and we average it throughout all samples. The results are shown in Table 2 and Table 3 (lower average entropy indicates sharper embedding distribution).

Table 2: The result of each-sample method.

| each-sample | FCL | BCL | BCL+FCL |
|---|---|---|---|
| top-1024 ↓ | 6.4713 | 5.7578 | 5.6576 |
| top-2048 ↓ | 6.6666 | 5.9306 | 5.8246 |

Table 3: The result of the mean-sample method.

| mean-sample | FCL | BCL | BCL+FCL |
|---|---|---|---|
| top-1024 ↓ | 6.1386 | 5.3328 | 5.1815 |
| top-2048 ↓ | 6.6666 | 5.9306 | 5.8246 |

We can observe that BCL+FCL achieves the smallest average entropy, which demonstrates the embedding distribution of BCL+FCL is sharper (Cf. the bottom right of Figure 1). Note that no contradiction arises between Observation 3.1 and Observation 3.2. The former reveals the connection between BCL and FCL through some theoretical approximations and assumptions, while the latter states that pushing negative pairs away in BCL and FCL can have complementary benefits. These consequences afford insights into a dual CL design which motivates our RecDCL.

## 4 THE RECDCL METHOD

Motivated by the analyses in Section 3, we develop a dual CL framework for recommendation, referred to as RecDCL. RecDCL is mainly characterized by two recommendation-fitted CL objectives: a Rec-FCL objective (Section 4.1) for driving the representations to be orthogonal, and a Rec-BCL objective (Section 4.2) for enhancing the robustness of the representations. Throughout this section, we use $\mathbf{E}_U \in \mathbb{R}^{B \times F}$ ($\mathbf{E}_I \in \mathbb{R}^{B \times F}$) to denote the user (item) embedding matrix, and $\mathbf{E}_U^{m,:}$ and $\mathbf{E}_U^{:,n}$ to respectively denote the $m$-th row and the $n$-th column of $\mathbf{E}_U$, where $B$ stands for the number of samples in a batch and $F$ represents the embedding dimension.

## 4.1 FCL Objective for Recommendation

**Eliminate redundancy between users and items.** To explore the alignment in an FCL way, we propose to extend the Barlow Twins objective function, namely UIBT, for self-supervised recommendations. More specially, we build a cross-correlation matrix computed from user and item embedding, and make it close to the identity matrix via invariance term and variance term as shown in Figure 2. Formally, the cross-correlation matrix between $\mathbf{E}_U$ and $\mathbf{E}_I$ can be computed as follows:

$$\mathbf{C}_{mn} = \frac{(\mathbf{E}_U^{:,m})^\top \mathbf{E}_I^{:,n}}{\|\mathbf{E}_U^{:,m}\| \|\mathbf{E}_I^{:,n}\|}, \tag{4}$$

where $1 \le m, n \le F$ denote the feature-wise indices of the embedding matrices. $\mathbf{C}$ is a square matrix that has the same dimensions

---

[2]The theoretical analysis for Observation 3.1 is provided in Appendix A.1.

[3]The theoretical and experimental analysis for Observation 3.2 is provided in Appendix A.2.

[4]Empirical evidence will be provided in Section 3.3.

**Figure 2: The overall framework of RecDCL.** In this framework, (a) denotes the cross-correlation matrix to an identity matrix between users and items in FCL; (b) and (c) stand for the distribution uniformity within users and items in FCL; (d) denotes the random distribution on user-item positive pairs in BCL; (e) demonstrates the final distribution on users and items produced by a dual CL.

as the encoder's output. Note that we define the invariance term and redundancy reduction term with a factor $1/F$ that scales the criterion as a function of the dimension:

$$\mathcal{L}_{UIBT} = \frac{1}{F} \underbrace{\sum_m (1 - C_{mm})^2}_{\text{invariance}} + \frac{\gamma}{F} \underbrace{\sum_m \sum_{m \neq n} C_{mn}^2}_{\text{redundancy reduction}} . \quad (5)$$

In Eq.5, the invariance term aims to make the diagonal elements of the matrix $C$ equal to 1 and meet the vector invariance to the distorted samples. Besides, the redundancy reduction term aims to make the off-diagonal elements of the matrix $C$ equal to 0 and reduce the redundancy within the output representation.

**Eliminate redundancy within users and items.** In addition, to further enhance the embedding diversity between different features. We propose a feature-wise uniformity optimization via a polynomial kernel, namely UUII. The polynomial kernel is the natural feature-wise way to present samples on the hypersphere. Considering the possible distribution gap between users and items, we calculate uniformity separately within the user embedding and the item embedding. The joint objective can be formulated as

$$\mathcal{L}_{UUII} = \frac{1}{2} \log \sum_{m \neq n} (a(\mathbf{E}_U^{:,m})^\top \mathbf{E}_U^{:,n} + c)^e + \frac{1}{2} \log \sum_{m \neq n} (a(\mathbf{E}_I^{:,m})^\top \mathbf{E}_I^{:,n} + c)^e, \quad (6)$$

where $a$, $c$ and $e$ are parameters of the polynomial kernel and set to $1$, $1e-7$ and $4$ by default, respectively. Note that feature-wise uniformity loss is only calculated via representations of in-batch samples since in-batch instances are more consistent with the actual user and item data distribution. As a result, the user/item distribution will be uniform, as shown in Figure 2. In addition, the exposure bias can be reduced by incorporating user/item distribution, which is illustrated in CLRec [49].

### 4.2 BCL Objective for Recommendation

Indeed, the proposed BCL objective and the proposed FCL objective (UIBT and UUII) are designed and evaluated for recommendations.

**Basic BCL.** To validate the generality, we propose a baseline that solely combines the basic BCL (DirectAU) and FCL (only

off-diagonal elements) methods. We directly design a summing loss function called DCL and conduct experiments on four public datasets. The summing loss function is described as:

$$\mathcal{L}_{DCL} = \mathcal{L}_{DirectAU} + \lambda \mathcal{L}_{FCL} \quad (7)$$

**Advanced BCL.** As analyzed in Section 3.2, combining the two dimensions FCL and BCL can intuitively benefit model learning. It motivates us to further improve optimization via a BCL objective. Generally, data augmentation can be achieved via a series of meaningful perturbations in many scenarios such as computer vision and neural language processing. However, positive user-item pairs in CF need to be preserved for representation invariance and are difficult to distort for data augmentation. To avoid this issue and achieve the same effect, as shown in Figure 2, we conduct data augmentation on output representation and generate contrastive but related views for representation learning, namely BCL. Owing to the simple design and effectiveness of LightGCN, we adopt it as the graph encoder $f_\theta$ to conduct node aggregation and propagation. After generating embedding of each layer and stacking multi-layer representations, we use historical embeddings [2, 8, 50] to perform augmentation on output embedding.

For the objective function, a trivial choice would be directly applying the original InfoNCE introduced in Eq. 2. A recent work [45] indicates that InfoNCE can be understood under a more general framework. It provides a perspective to unify InfoNCE with another popular SSL method SimSiam, which implicitly performs CL through the stop-gradient and asymmetric trick. We empirically find that this implicit CL design achieves better performance and choose it as our implementation.

Suppose $\mathbf{E}_U$ ($\mathbf{E}_I$) is the current embedding generate by the encoder $f_\theta$ and $\mathbf{E}_U^{(h)}$ ($\mathbf{E}_I^{(h)}$) is the historical embedding. The perturbed representation $\hat{\mathbf{E}}_U$ is calculated by combining $\mathbf{E}_U^{(h)}$ and $\mathbf{E}_U^{(h)}$:

$$\hat{\mathbf{E}}_U = \tau \mathbf{E}_U^{(h)} + (1 - \tau)\mathbf{E}_U, \quad \hat{\mathbf{E}}_I = \tau \mathbf{E}_I^{(h)} + (1 - \tau)\mathbf{E}_I, \quad (8)$$

where $\tau$ is a hyper-parameter that controls the embedding information preservation ratio from a prior training iteration. Note that we perform representation distortion on the historical embedding from prior training iterations in the target network instead of perturbing

the input commonly [22] or the current node embedding [41, 42] directly as used in previous CL-based recommendation methods.

Besides, online and target networks share the same graph encoder $f_\theta$ in this component, which can reduce the additional memory and computation cost. Therefore, the optimization of batch-wise output augmentation can be formulated as follows:

$$\mathcal{L}_{BCL} = \frac{1}{2}S(h(\mathbf{E}_U), sg(\hat{\mathbf{E}}_I)) + \frac{1}{2}S(sg(\hat{\mathbf{E}}_U), h(\mathbf{E}_I)), \qquad (9)$$

where $h(\cdot)$ is a multi-layer perceptron network; $sg(\cdot)$ is the stop-gradient operation; and $S(\cdot, \cdot)$ denotes the cosine distance.

## 4.3 Objective and Training

In a word, the main goal of RecDCL is to design an FCL objective and an advanced BCL objective that capture embedding importance and maximize the benefits from a dual CL to encourage embedding learning, as shown in Figure 2. In practice, we jointly optimize these three objectives with trade-off hyperparameters $\alpha$ and $\beta$ as follows:

$$\mathcal{L} = \mathcal{L}_{UIBT} + \alpha\mathcal{L}_{UUII} + \beta\mathcal{L}_{BCL}. \qquad (10)$$

Finally, we calculate the ranking score function by using the inner product between user and item representations.

**Training Algorithm.** Let $|\mathcal{U}|$, $|\mathcal{I}|$ and $|\mathcal{E}|$ denote the number of user nodes, item nodes, and edges in a user-item bipartite graph $\mathcal{G}$, respectively. $B$ and $F$ stand for the batch size and the feature size. $s$ represents the number of epochs and $L$ is the number of GCN layers. The training process is shown in Algorithm 1 and the time complexity is analyzed in Appendix C.2.

---

**Algorithm 1:** The training process of the RecDCL.

**Input:** $f_\theta$: graph encoder, $L$: number of layer.
**Output:** encoder parameters $\theta$.
**Data:** $\mathcal{G}$: user-item bipartite graph, $\mathcal{U}$: user set, $\mathcal{I}$: item set,
$\qquad\mathcal{R}$: positive pairs
**for** *each mini-batch with positive pairs* $(u, i) \in \mathcal{R}$ **do**
$\qquad$ Initialize $\mathbf{e}_u^{(0)}, \mathbf{e}_i^{(0)}, \forall u \in \mathcal{U}, \forall i \in \mathcal{I}$;
$\qquad$ Generate $\mathbf{e}_u$ and $\mathbf{e}_i$ via encoder $f_\theta(u, L)$ and $f_\theta(i, L)$;
$\qquad$ Normalize $\mathbf{e}_u$: $\mathbf{e}_u = \mathbf{e}_u/\|\mathbf{e}_u\|$;
$\qquad$ Normalize $\mathbf{e}_i$: $\mathbf{e}_i = \mathbf{e}_i/\|\mathbf{e}_i\|$;
$\qquad$ Calculate UIBT loss by Eq. 5;
$\qquad$ Calculate UUII loss by Eq. 6;
$\qquad$ Calculate AUG loss by Eq. 9;
$\qquad$ Calculate total loss by Eq. 10.

---

## 5 EXPERIMENTS

In this section, we conduct extensive experiments on four public datasets and one industrial dataset in real-world application to validate the effectiveness of RecDCL. We introduce the overall performance and study the influence of each design of RecDCL. We also describe experimental settings, analyze the efficiency, and give detailed hyper-parameter sensitivity analysis in Appendix C.

## 5.1 Overall Performance

In Table 4, we show the overall top-20 performance of all the baselines and our RecDCL. Specifically, we highlight the best result in **bold** and the second best result in underline, calculate the relative

improvement (%Improv.) for our methods, and show the $p$-value in t-test experiments on each dataset. We give detailed observations based on these experimental results.

**Comparison with MF-based models.** From Table 4, we can observe that SSL-based models except BUIR on Game, particularly our RecDCL, can obtain superior results on most conditions. This demonstrates that SSL-based models have remarkable advantages in the problem of extremely sparsity data. More specifically, BPR-MF outperforms VAE-based models and GNNs-based models on Beauty and Food datasets, while the performance of NeuMF is the poorest on all datasets. This indicates that the VAE-based models and GNNs-based models can better capture the interactions between users and items via variational autoencoders and graph neural networks.

**Comparison with VAE-based models.** Compared to GNNs-based models, VAE-based models obtain inferior results on Beauty and Game even though they are effective on Yelp. In contrast, our RecDCL as well as contrastive learning-based DirectAU are robust and achieve significantly better on all four datasets. This suggests that SSL-based models represented by our RecDCL can address the sample distribution of users and items in user-item interaction well.

**Comparison with GNNs-based models.** As shown in Table 4, SSL-based models (CLRec and our RecDCL) are higher than GNNs-based models in terms of Recall@20 and NDCG@20 on all four datasets, while BUIR performs well on Beauty and Food datasets. Especially, our proposed RecDCL exceeds the state-of-the-art GNNs-based model LightGCN by 17.06%, 17.87%, 6.46%, and 37.49% in terms of Recall@20 on all four datasets respectively, which verifies the effectiveness of constructing SSL-based loss function instead of BPR loss with negative sampling. Moreover, this advantage also answers the question mentioned in Section 1 that CF-based recommendation models without negative sampling can still be effective.

**Comparison with SSL-based models.** Particularly, in Table 4, we show existing batch-wise CL objectives based BUIR, CLRec, and DirectAU in SSL-based recommendation outperforms the state-of-the-art GNNs-based methods in CF, although only using positive user-item pairs to construct contrastive learning loss function. Particularly, our proposed RecDCL consistently derives promising results on all four datasets and improves by 1.33%-5.20% and 3.35%-5.34% in terms of Recall@20, and NDCG@20, which verifies the effectiveness of incorporating the feature-wise objectives for self-supervised recommendation in RecDCL. It also meets the viewpoint in Section 1 that feature-wise CL is worth considering and the theoretical analysis in Section 3 about the effective combination of feature-wise objectives and batch-wise objectives for SSR.

## 5.2 Study of RecDCL

To investigate the reasons for RecDCL's effectiveness, we perform comprehensive ablation experiments to study the necessity of each component in RecDCL. Considering the limited space, we only show the analysis on representative datasets (Beauty and Yelp). The result analysis on Food and Game datasets is represented in Table 10 of Appendix C.5.

**Effect of feature-wise objective UIBT.** As discussed in Section 4, UIBT achieves approximate results compared to the base encoder

**Table 4: Overall top-20 performance (% is omitted) comparison with representative models on four datasets (R and N are the abbreviations for Recall and NDCG).**

| Models | Dataset | Beauty | | Food | | Game | | Yelp | |
|---|---|---|---|---|---|---|---|---|---|
| | Metrics | R@20 | N@20 | R@20 | N@20 | R@20 | N@20 | R@20 | N@20 |
| Base | Pop | 3.25 | 1.31 | 5.74 | 3.40 | 2.82 | 1.06 | 1.58 | 0.96 |
| MF-based | BPR-MF | 14.12 | 6.62 | 27.02 | 21.04 | 18.16 | 8.33 | 6.92 | 4.29 |
| | NeuMF | 7.66 | 3.46 | 15.28 | 8.79 | 10.36 | 4.28 | 6.01 | 3.63 |
| VAE-based | Mult-VAE | 11.37 | 5.46 | 24.89 | 20.77 | 15.50 | 7.18 | 9.51 | 5.84 |
| | RecVAE | 12.76 | 6.37 | 26.69 | 22.29 | 17.65 | 8.38 | 10.70 | 6.69 |
| GNNs-based | NGCF | 13.27 | 6.28 | 26.84 | 20.96 | 18.04 | 8.31 | 7.29 | 4.45 |
| | LightGCN | 13.48 | 6.25 | 24.56 | 16.77 | 19.20 | 8.91 | 8.43 | 5.23 |
| SSL-based | BUIR | 14.60 | 7.29 | 28.26 | 22.19 | 15.04 | 6.73 | 8.08 | 4.97 |
| | CLRec | 15.17 | _7.56_ | 27.64 | 20.65 | 20.12 | 9.60 | 10.95 | 6.89 |
| | DirectAU | _15.43_ | 7.49 | _28.57_ | _22.41_ | _20.14_ | _9.55_ | _10.97_ | _6.92_ |
| | **DCL** | **15.59** | 7.54 | **28.63** | **22.52** | **20.20** | **9.58** | **10.99** | **6.96** |
| | %Improv. | 1.04% | 0.67% | 0.21% | 0.49% | 0.30% | 0.31% | 0.18% | 0.58% |
| | **RecDCL** | **15.78** | **7.89** | **28.95** | **23.27** | **20.44** | **9.87** | **11.59** | **7.28** |
| | %Improv. | 2.27% | 5.34% | 1.33% | 3.84% | 1.49% | 3.35% | 5.65% | 5.20% |
| | *p*-value | 0.004115 | 0.000478 | 0.002255 | 0.000017 | 0.264695 | 0.029848 | 0.001402 | 0.006503 |

[1] Note that we tune embedding size from 32 to 2048 and report the best results for all baselines and our method RecDCL. Generally, the embedding size is set by default to 64.

[2] Indeed, RecDCL is the very first work to explore the effectiveness of FCL for recommendations. We have looked and found that there are no appropriate baselines for FCL. To comprehensively compare, we conduct the experiments in ablation studies, that is UIBT for FCL.

model LightGCN in feature-wise objectives. Table 5 shows the comparison between LightGCN and the components of UIBT on Beauty and Yelp. We can find that (1) only "w/ UIBT" is better than LightGCN on Beauty in terms of Recall@20 and NDCG@20/ (2) To validate the effectiveness of UIBT, we compare results between "w/ UUII" and "w/ UIBT & UUII". From Table 5, we find that the latter value is not just higher than LightGCN, and higher than "w/ UUII" on Beauty and Yelp. (3) Similarly, the comparison between "w/ BCL" and "w/ UIBT & BCL" also shows this ascending trend on two datasets, which again demonstrates the importance of interactions between users and items either in FCL or BCL.

**Table 5: Performance comparison of different designs of RecDCL on Beauty and Yelp.**

| Method | Beauty | | Yelp | |
|---|---|---|---|---|
| | R@20 | N@20 | R@20 | N@20 |
| LightGCN | 13.48 | 6.25 | 8.43 | 5.28 |
| w/ UIBT | 14.78 | 7.47 | 9.92 | 6.18 |
| w/ UUII | 1.01 | 0.50 | 0.06 | 0.03 |
| w/ BCL | 14.90 | 7.51 | 10.08 | 6.36 |
| w/ UIBT & UUII | 14.88 | 7.43 | _11.00_ | _6.85_ |
| w/ UIBT & BCL | _15.64_ | _7.63_ | 10.73 | 6.78 |
| w/ UUII & BCL | 15.16 | 7.59 | 7.65 | 4.66 |
| RecDCL | **15.78** | **7.89** | **11.59** | **7.28** |
| %Improv. | 17.06% | 26.24% | 37.49% | 37.88% |

**Effect of feature-wise objective UUII.** As shown in Table 5, (1) Compared to "w/ UIBT" and "w/ BCL", the performance of only

feature-wise UUII, namely "w/ UUII", is extremely poor on Beauty and Yelp. This finding is similar to DirectAU which only optimizes uniformity, which just goes to show the importance of optimizing uniformity. (2) Compared to "w/ UIBT" and "w/ UIBT & UUII", the performance of "w/ UIBT & UUII" consistently outperforms "w/ UIBT" on Beauty and Yelp datasets. This shows the importance of sample distribution within users and items for recommendation. Besides, addressing feature-wise interaction distribution and user (item) data distribution is vital for performance improvements. (3) The comparison between "w/ BCL" and "w/ UUII & BCL" shows the performance improvement given by "w/ UUII". (4) In fact, the UUII component simulates the real distribution among users (items) and meets the essential characteristic of contrastive learning. In combination with the above analysis, the UIBT maximizes the similar representation while minimizing the dissimilar embedding between users and items of in-batch, and the UUII nomial minimizes the similarity among users (items) in a feature-wise way.

**Effect of batch-wise augmentation BCL.** From Table 5, we have some observations: (1) we first conduct experiments solely on batch-wise augmentation and find that it is better than base LightGCN on the Beauty dataset, which shows the benefits of data augmentation that reinforces the interaction between users and items. (2) To further study the influence of output augmentation, we compare "w/ UUII" and "w/ UUII & BCL", which demonstrates the effectiveness of BCL. Note that the performance of "w/ UUII & BCL" is better than "w/ BCL" even if the performance of only "w/ UUII" is poor, which again shows the importance of combining interactions with user (item) distribution. (3) Meanwhile, the comparison between "w/ UIBT" and "w/ UIBT & BCL" also shows the effectiveness of BCL on these two datasets.

## 5.3 Industrial Results

To further study the effectiveness of our method, we employ RecDCL on an industrial dataset, which is collected from an online payment platform. We first extract links between users and retailers from user payment behavior logs in a month, December 2022. Then, we define users who conduct more than 3 payments as core users and subsample the users from the core user set. The general statistics of this dataset are summarized in Table 6.

**Table 6: Statistics of payment dataset.**

| Dataset | # Users | # Retailers | # Inter | Density | Sparsity |
|---------|---------|-------------|---------|---------|----------|
| Payment | 83,810  | 23,318      | 797,828 | 0.0408% | 99.96%   |

**Table 7: Performance of RecDCL on payment platform data.**

| Method    | BPR-MF | Mult-VAE | LightGCN | DirectAU | RecDCL | %Improv. |
|-----------|--------|----------|----------|----------|--------|----------|
| Recall@20 | 35.27  | 31.87    | 33.48    | 31.34    | **36.47** | 3.40%    |
| NDCG@20   | 16.61  | 15.19    | 15.08    | 14.07    | **17.86** | 7.53%    |

We conducted an offline experiment on a worker equipped with NVIDIA-A100 GPU, and 40GB memory. Following the experimental setting in Section 5, the users' behaviors are also split into training/validation/test sets with a ratio of 0.8/0.1/0.1. The main task focuses on retrieving top-$N$ business from the item pool and then recommending them to users. To evaluate the task, we introduce Recall@20 and NDCG@20 as metrics, which are carefully considered in the real industrial platform.

Notice that although the baseline of this platform is BPR-MF, we also conduct experiments on LightGCN and DirectAU to further compare our method with others. As shown in Table 4, the overall effectiveness of DirectAU is better on the private dataset than on the public datasets. Regarding this, we speculate that the reason behind this is the long-tail effect of this task is more serious, and DirectAU has a uniform loss, which may be a little bad in the case of such a serious long-tail effect. Table 7 demonstrates that our method performs significantly compared with other baselines in terms of two metrics Recall@20 and NDCG@20, improving by 3.40% and 7.53% respectively.

## 6 RELATED WORK

### 6.1 Collaborative Filtering

Collaborative filtering plays a vital role in modern recommendation systems [30]. The key idea of CF is that similar users tend to have similar preferences on items. To address the issue of data sparsity in CF, Matrix Factorization (MF) decomposes the original user-item interacted matrix to the low-dimensional matrix with user (item) latent features. NeuMF [15] based on a deep neural network is proposed to learn rich information and compress latent features. Furthermore, based on the viewpoint that the modeling capacity of linear factor models often is limited, Liang et al. [23] introduce the generative model - variational autoencoders (VAEs) with multinomial likelihood into collaborative filtering tasks and propose Mult-VAE, which shows excellent performance for top-$N$ recommendations mainly owing to well modeling user-item implicit feedback data. Afterward, RecVAE [31] improves several aspects of

the Mult-VAE, mainly including a novel encoder, novel approach to setting hyperparameter, and new training strategy, and then performs well.

With the development of graph neural networks (GNNs), GNNs-based CF models are widely studied and proposed. The observed user-item interaction matrix can be built as a bipartite graph, and GNNs-based models [3, 14, 36] aggregate information from neighbors and capture high-order connection information on graph-structure data. Take LightGCN [14] as an example, it simplifies the GNN architecture by removing nonlinear activation as well as feature transformation based on NGCF [36]. Such methods do produce relatively ideal results, however, there still remains a tricky challenge when we are faced with extremely sparse data.

### 6.2 Contrastive Learning for Recommendation

Recently, self-supervised learning for recommendation has attracted increasing attention and a number of SSL-based CF [22, 24, 33, 37, 39, 49, 50] has achieved competitive results. According to the learning objective and negative sampling strategy, we divide existing SSL-based CF into two categories: **(1) With sampling:** the state-of-the-art model SimGCL [42] perturbed the current representation of each node via random and different noise-based data augmentation. **(2) Without sampling:** inspired by MoCo [13], BUIR [22] proposes an asymmetric structure to learn user and item representation solely on user-item positive pairs. SelfCF [50] inherits the Siamese network structure of SimSiam's architecture [5] and optimizes Cosine Similarity loss on output augmentation obtained in in-batch positive-only data. CLRec [49] adopts InfoNCE loss to tackle exposure bias for recommendation. DirectAU [33] focuses on the desired properties of representations from the alignment and uniformity metrics and optimizes these two properties via a new loss function in CF. However, none of these works considered CL-based recommendations from the perspective of feature-wise, resulting in limited performance. Barlow Twins [44] is the first to investigate the feature-wise objective in CV domains. Motivated by Barlow Twins, we study the feasibility of feature-wise objectives and design a dual CL with BCL and FCL objectives. Due to the prominent results on SSL-based models without sampling, we follow this design principle to optimize the learning objective.

## 7 CONCLUSION

In this work, we theoretically reveal the connection of the objective between BCL and FCL and show a cooperative benefit by using them both, which motivates us to develop a dual CL called RecDCL that jointly optimizes BCL and FCL training objectives to learn informative representations for recommendation. Then we investigate the two desired properties of representation — alignment and uniformity — in FCL objectives for CF, and perform data augmentation on output vectors for robustness in BCL objective. Extensive experiments on four public datasets and an industrial dataset show the superiority of our proposal considering the FCL objectives. We hope RecDCL could attract the CF community's attention to the learning paradigm toward FCL perspective representation properties. In the future, we will investigate other CL-based training objectives that also favor feature-wise perspectives to improve effectiveness and efficiency.

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

## A MORE DETAILED EXPLANATION

### A.1 Connection Between BCL and FCL

The part provides the theoretical analyses on Observation 3.1 in the main text, which is restated as follows for convenience.

OBSERVATION A.1. *If the two embedding matrices are standardized (i.e., they have mean zero and standard deviation one), then the objectives of BCL and FCL can be approximately transformed to each other.*

Our main idea is to transform both BCL and FCL objectives into matrix forms and then connect them through the algebraic properties of matrices.

Formally, let $\mathbf{Z} \in \mathbb{R}^{N \times D}$ and $\hat{\mathbf{Z}} \in \mathbb{R}^{N \times D}$ denote the embedding matrices of raw samples and the corresponding augmented samples in a batch, respectively. The BCL objective can be formulated as:

$$-\sum_{i=1}^{N} \log \left( \frac{\exp\left(\mathbf{z}_i^\top \hat{\mathbf{z}}_i\right)}{\sum_{j=1}^{N} \exp\left(\mathbf{z}_i^\top \hat{\mathbf{z}}_j\right)} \right) = \sum_{i=1}^{N} \underbrace{-\mathbf{z}_i^\top \hat{\mathbf{z}}_i}_{\text{draw close}} + \underbrace{\log \sum_{j=1}^{N} \exp\left(\mathbf{z}_i^\top \hat{\mathbf{z}}_j\right)}_{\text{push away}}. \tag{11}$$

Here, we assume that $\mathbf{z}_i^\top \hat{\mathbf{z}}_j \in [0,1]$ for any $1 \leq i, j \leq N$, as the negative terms contribute little to the log-sum-exponential function. With this assumption, let us consider a surrogate of R.H.S. in Eq. 11:

$$\sum_{i=1}^{N} \underbrace{(1 - \mathbf{z}_i^\top \hat{\mathbf{z}}_i)^2}_{\text{draw close}} + \underbrace{\sum_{j \neq i} (\mathbf{z}_i^\top \hat{\mathbf{z}}_j)^2}_{\text{push away}}, \tag{12}$$

where the push-away term is a reasonable approximation to the counterpart in Eq. 11 since both the sum-square function and the log-sum-exponential function relatively enhance larger terms while weakening smaller terms for non-negative values. With this surrogate, we can transform the BCL objective into a matrix form:

$$\sum_{i=1}^{N} (1 - \mathbf{z}_i^\top \hat{\mathbf{z}}_i)^2 + \sum_{i=1}^{N} \sum_{j \neq i} (\mathbf{z}_i^\top \hat{\mathbf{z}}_j)^2 \quad \text{(BCL objective surrogate)}$$
$$= \|\mathbf{I} - \mathbf{Z}\hat{\mathbf{Z}}^\top\|_F^2 = N + \text{Tr}(\hat{\mathbf{Z}}\mathbf{Z}^\top \mathbf{Z}\hat{\mathbf{Z}}^\top) - \text{Tr}(\hat{\mathbf{Z}}\mathbf{Z}^\top) - \text{Tr}(\mathbf{Z}\hat{\mathbf{Z}}^\top). \tag{13}$$

On the other hand, with the condition that both $\mathbf{Z}$ and $\hat{\mathbf{Z}}$ are standardized, setting the redundancy reduction weight $\lambda = 1$ in the FCL objective yields

$$\sum_{i=1}^{D} (1 - C_{ii})^2 + \sum_{i=1}^{D} \sum_{j \neq i} (C_{ij})^2 \quad \text{(FCL objective)}$$
$$= \|\mathbf{I} - \mathbf{Z}^\top \hat{\mathbf{Z}}\|_F^2 = D + \text{Tr}(\hat{\mathbf{Z}}^\top \mathbf{Z} \mathbf{Z}^\top \hat{\mathbf{Z}}) - \text{Tr}(\hat{\mathbf{Z}}^\top \mathbf{Z}) - \text{Tr}(\mathbf{Z}^\top \hat{\mathbf{Z}}). \tag{14}$$

Comparing Eq. 13 and Eq. 14, we find that they differ by only a constant $|N - D|$ if $\mathbf{Z}^\top \hat{\mathbf{Z}} = \hat{\mathbf{Z}}^\top \mathbf{Z}$ and $\mathbf{Z}\hat{\mathbf{Z}}^\top = \hat{\mathbf{Z}}\mathbf{Z}^\top$ [5], i.e.,

$$\text{Tr}(\hat{\mathbf{Z}}\mathbf{Z}^\top \mathbf{Z}\hat{\mathbf{Z}}^\top) - \text{Tr}(\hat{\mathbf{Z}}\mathbf{Z}^\top) - \text{Tr}(\mathbf{Z}\hat{\mathbf{Z}}^\top)$$
$$= \text{Tr}(\hat{\mathbf{Z}}\mathbf{Z}^\top \mathbf{Z}\hat{\mathbf{Z}}^\top) - \sum_{i=1}^{N} \lambda_i(\hat{\mathbf{Z}}^\top \mathbf{Z}) - \sum_{i=1}^{N} \lambda_i(\mathbf{Z}^\top \hat{\mathbf{Z}})$$
$$= \text{Tr}(\hat{\mathbf{Z}}^\top \mathbf{Z} \mathbf{Z}^\top \hat{\mathbf{Z}}) - \sum_{i=1}^{D} \lambda_i(\hat{\mathbf{Z}}^\top \mathbf{Z}) - \sum_{i=1}^{D} \lambda_i(\mathbf{Z}^\top \hat{\mathbf{Z}})$$
$$= \text{Tr}(\hat{\mathbf{Z}}^\top \mathbf{Z} \mathbf{Z}^\top \hat{\mathbf{Z}}) - \text{Tr}(\hat{\mathbf{Z}}^\top \mathbf{Z}) - \text{Tr}(\mathbf{Z}^\top \hat{\mathbf{Z}}), \tag{15}$$

where $\lambda_i$ stands for the $i$-th eigenvalue, and the equation from the second row to the third row holds because $\hat{\mathbf{Z}}^\top \mathbf{Z}, \mathbf{Z}^\top \hat{\mathbf{Z}}, \hat{\mathbf{Z}}^\top \mathbf{Z}$ and $\mathbf{Z}^\top \hat{\mathbf{Z}}$ share exactly the same non-zero eigenvalues [16]. Note that the conditions $\mathbf{Z}^\top \hat{\mathbf{Z}} = \hat{\mathbf{Z}}^\top \mathbf{Z}$ and $\mathbf{Z}\hat{\mathbf{Z}}^\top = \hat{\mathbf{Z}}\mathbf{Z}^\top$ can be satisfied if $\mathbf{Z} \approx \hat{\mathbf{Z}}$, which is mild requirement since the objective of CL naturally leads to $\mathbf{Z} \approx \hat{\mathbf{Z}}$. Eq. 15 reveals that, under the assumed conditions, the objectives of BCL and FCL can be approximately transformed to each other.

### A.2 Joint BCL and FCL

OBSERVATION A.2. *For normalized sample embedding, pushing negative pairs away has different influences to embedding learning between BCL and FCL. For BCL, it encourages samples to be evenly distributed in the embedding space. For FCL, it tends to drive the representations of samples to be orthogonal. This difference is mainly due to that BCL encourages the inner product of negative pairs (in the batch dimension) to be as small as possible; but FCL only enforces the inner product of negative pairs (in the feature dimension) to be close to zero, which implicitly encourages the representations of samples (in the batch dimension) to be orthogonal. If we combine BCL and FCL, pushing negative pairs away will not only encourage sample representations to be evenly distributed in the embedding space but also help eliminate redundant solutions (Cf. Figure 1). This regularity can benefit embedding learning as the embedding dimension increases[6].*

The key to our interpretation for Observation 3.2 is to consider the quality of the solution sets resulting from the push-away objectives (for negative pairs) in BCL and FCL: using only BCL or only FCL would lead to a large number of redundant solutions while combining BCL and FCL would effectively reduce this redundancy and never miss an optimal solution. We start with the special case of one negative pair (the example shown in Figure 1), followed by a general case of an arbitrary number of negative samples.

**Theory-wise.** Suppose that $\mathbf{z}_1 \in \mathbb{R}^D$ and $\mathbf{z}_2 \in \mathbb{R}^D$ are representations of a negative pair residing on a hyper-sphere surface $\mathcal{S}^{D-1} \triangleq \{\mathbf{z} \in \mathbb{R}^D \mid \|\mathbf{z}\|_2 = 1\}$. Then the push-away objective in BCL and FCL can be respectively formulated as

$$\min_{\mathbf{z}_1, \mathbf{z}_2 \in \mathcal{S}^{D-1}} \underbrace{\exp(\mathbf{z}_1^\top \mathbf{z}_2)}_{\text{push-away in BCL}} \triangleq \min_{\mathbf{z}_1, \mathbf{z}_2 \in \mathcal{S}^{D-1}} \mathbf{z}_1^\top \mathbf{z}_2, \tag{16}$$

and

---

[5]These two conditions imply that $\text{Tr}(\hat{\mathbf{Z}}\mathbf{Z}^\top \mathbf{Z}\hat{\mathbf{Z}}^\top) = \text{Tr}(\hat{\mathbf{Z}}^\top \mathbf{Z} \mathbf{Z}^\top \hat{\mathbf{Z}})$, and all $\hat{\mathbf{Z}}^\top \mathbf{Z}, \mathbf{Z}^\top \hat{\mathbf{Z}}$, $\hat{\mathbf{Z}}^\top \mathbf{Z}$ and $\mathbf{Z}^\top \hat{\mathbf{Z}}$ can be similarly diagonalized.
[6]Empirical evidence will be provided in Section C.6.

$$\min_{\mathbf{z}_1,\mathbf{z}_2 \in \mathcal{S}^{D-1}} \underbrace{\sum_{i=1}^{D}\sum_{j\neq i}(C_{ij})^2}_{\text{push-away in FCL}} \triangleq \min_{\mathbf{z}_1,\mathbf{z}_2 \in \mathcal{S}^{D-1}} ||\text{off-diag}(\mathbf{z}_1\mathbf{z}_1^\top + \mathbf{z}_2\mathbf{z}_2^\top)||_F^2,$$

(17)

where off-diag$(\cdot)$ denotes a projection operator which preserves the off-diagonal elements of a matrix. It can be verified that the optimal solution to (16) and (17) is given by

$$\mathcal{Z}_B^\star = \{(\mathbf{z}_1, \mathbf{z}_2) \mid \mathbf{z}_1, \mathbf{z}_2 \in \mathcal{S}^{D-1}, \mathbf{z}_1 = -\mathbf{z}_2\} \qquad (18)$$

and

$$\mathcal{Z}_F^\star = \{(\mathbf{z}_1, \mathbf{z}_2) \mid \mathbf{z}_1, \mathbf{z}_2 \in \mathcal{S}^{D-1}, \text{off-diag}(\mathbf{z}_1\mathbf{z}_1^\top + \mathbf{z}_2\mathbf{z}_2^\top) = \mathbf{0}\}, \quad (19)$$

respectively. In general, a solution in $\mathcal{Z}_B^\star$ (cf. the top right of Figure 1) does not admit a solution in $\mathcal{Z}_F^\star$ (cf. the bottom left of Figure 1). However, if $\mathbf{z}_1$ is a one-hot vector and $\mathbf{z}_2 = -\mathbf{z}_1$ (cf. the bottom right of Figure 1), substituting them into (16) and (17) respectively yields the optimal value -1 and 0, implying that they exactly fall into the joint solution set $\mathcal{Z}_B^\star \cap \mathcal{Z}_F^\star \neq \emptyset$. The above analysis explains the derivation of Figure 1 in our paper.

Regarding "implicitly encourages the representations of samples (in the batch dimension) to be orthogonal", let us first consider a special case where the normalized embedding matrices $\mathbf{Z}, \hat{\mathbf{Z}} \in \mathbb{R}^{B \times F}$ and the feature dimension $B$ equals the batch dimension $F$. In this case, the FCL objective which optimize $\mathbf{Z}^\top \hat{\mathbf{Z}}$ to approach the identity matrix $\mathbf{I}$ indeed encourages the representations of samples to be orthogonal (i.e., $\mathbf{Z}\hat{\mathbf{Z}}^\top \approx \mathbf{I}$) since $\mathbf{Z}\hat{\mathbf{Z}}^\top = \mathbf{I} \iff \mathbf{Z}^\top \hat{\mathbf{Z}} = \mathbf{I}$. For the case $B > F$, FCL actually encourages feature embeddings to fall into $F$ clusters which are orthogonal to each other. For the case $B < F$, FCL encourages feature embeddings to be orthogonal in a subspace.

Furthermore, for the general case where $N$ negative samples constitute a representation matrix $\mathbf{Z} \in \mathbb{R}^{N \times D}$, the push-away objective in BCL and FCL can be respectively formulated as

$$f_B(\mathbf{Z}) \triangleq \mathbf{1}^\top \text{off-diag}(\mathbf{Z}\mathbf{Z}^\top)\mathbf{1} \text{ and } f_F(\mathbf{Z}) \triangleq ||\text{off-diag}(\mathbf{Z}^\top\mathbf{Z})||_F^2, \quad (20)$$

where $\mathbf{1} \in \mathbb{R}^N$ denotes a all-one vector. It is easy to verify that $f_B$ and $f_F$ are respectively invariant under the right rotation and the left rotation, i.e.,

$$f_B(\mathbf{Z}\mathbf{R}_B) = f_B(\mathbf{Z}) \text{ and } f_F(\mathbf{R}_F\mathbf{Z}) = f_F(\mathbf{Z}), \qquad (21)$$

where $\mathbf{R}_B \in \mathbb{R}^{D \times D}$ and $\mathbf{R}_F \in \mathbb{R}^{N \times N}$ denote any rotation matrices. FCL inherently tends to make each feature dimension orthogonal to each other. However, this does not necessarily imply that the samples are positioned on the axes. These rotation-invariances induce redundancy when using BCL only or FCL only. Positioning the samples on the axes represents merely one specific instance of many redundant solutions. In contrast, combing BCL and FCL eliminates this redundancy since $f_{B+F}(\mathbf{Z}) \triangleq f_B(\mathbf{Z}) + f_F(\mathbf{Z})$ is neither left-rotation invariant nor right-rotation invariant in general. On the other hand, one can construct an optimal solution to $f_{B+F}$ that also admits the optimality of both $f_B$ and $f_F$, so using $f_{B+F}$ never miss the optimal solution.

In summary, combining the push-away objectives in BCL and FCL would reduce redundant solutions but never miss an optimal solution, thus qualifying as a more reasonable regularization. These analyses support our claims in Observation 3.2.

**Experiment-wise.** Remarkably, in the theoretical part, we consider the case where the vanilla BCL and FCL are directly combined by means of addition for easy of analysis. In practice, RecDCL incorporates more advanced BCL and FCL techniques in a soft way. This creates a slight discrepancy between theory and experience: the gap in average entropy between RecDCL and BCL is not very obvious. Nevertheless, they are consistent in trend.

## B MORE DETAILED DISCUSSION

As for the relation and differences between batch-wise and feature-wise CL, we will perform more detailed discussion.

**Relation.** As formulated in Eq. 3, the objective function of Barlow Twins essentially plays a role similar to the contrastive term in existing objective functions for self-supervised learning, such as the representative InfoNCE in SGL [37]. In this regard, we have analyzed it theoretically in A.1.

**Differences.** Compared to batch-wise CL methods, there exist certain advantages in our method or other batch-wise CL methods due to important conceptual differences between batch-wise and feature-wise CL. Through a detailed analysis, the main differences between batch-wise and feature-wise CL focus on two aspects.
• **FCL** methods represented by Barlow Twins strongly benefit from pretty high-dimensional embeddings, which still is acceptable although a rather high computational cost.
• **BCL**, intriguingly, the batch-wise objective maximizes the variability of the embedding vectors via maximizing the pairwise distance between all interacted user-item pairs, while feature-wise CL methods do this by decorrelating each component of these embedding vectors. Thus, to provide better performance of recommendation, we develop RecDCL by combining the strengths of feature-wise and batch-wise objectives which can maximize the benefits of CL.
• **FCL+BCL (CL4CTR)** focus is on addressing the "long tail" distribution of feature frequencies by directly learning accurate feature representations through BCL and FCL. RecDCL focuses on combining the FCL objective and BCL objective whose effectiveness to reduce redundant solutions but never miss an optimal solution, making it a more reasonable regularization technique.

## C MORE DETAILED EXPERIMENTS

### C.1 Experimental Settings

**Baselines.** We compare various representative baselines including pop model, MF-based models (BPR-MF and NeuMF), VAE-based models (Mult-VAE and RecVAE), GNNs-based models (NGCF and LightGCN), and SSL-based models (BUIR, CLRec, and DirectAU). The detailed description is presented as follows:
• Pop recommends the most popular items to each user.

- BPR-MF [29] optimizes MF with Bayesian personalized ranking (BPR) loss by sampling negatives of interactions between users and items.
- NeuMF [15] utilizes a multi-layer perceptron to model users and items based on implicit feedbacks.
- Mult-VAE [23] is a typical item-based CF method based on variational autoencoders (VAEs) to maximum entropy discrimination.
- RecVAE [31] is also based on VAEs and improves the performance of Mult-VAE [23] by reconstructing user representations.
- NGCF [36] proposes a message-passing framework and performs graph convolution for collaborative filtering in first-order and high-order propagation way.
- LightGCN [14] is the state-of-the-art GNNs-based model, which removes the activation function and feature transformation of NGCF to simplify the design of graph convolution for recommendation.
- BUIR [22] is the first work that only uses user-item positive pair without any negative sampling in contrastive learning for collaborative filtering. Inspired by SimSiam [5], BUIR trains online encoder and target encoder for one-class collaborative filtering.
- CLRec [49] addresses exposure bias via InfoNCE loss for recommendation.
- DirectAU [33] focuses on the representation of users and items obtained from representation learning. It considers the alignment of user-item positive pairs to minimize the distance of positive pairs and calculates the uniform metric between users and items to maximize the distance between dissimilar users.

**Datasets.** We adopt four widely public datasets with different scales and one real-world dataset, in which the specific information of them are described in Table 8 and Table 6, respectively.

**Table 8: Statistics of datasets.**

| Dataset | Beauty | Food | Game | Yelp |
|---------|--------|------|------|------|
| #Users | 22,364 | 127,279 | 37,419 | 31,669 |
| #Items | 12,102 | 40,995 | 14,079 | 38,049 |
| #Inter | 198,502 | 1,141,946 | 343,481 | 1,561,406 |
| Avg/user | 8.88 | 8.97 | 9.18 | 49.31 |
| Avg/item | 16.4 | 27.86 | 24.4 | 41.04 |
| Density | 0.0733% | 0.0219% | 0.0652% | 0.1296% |
| Sparsity | 99.93% | 99.98% | 99.93% | 99.87% |

- Beauty[7] is one of the types collected from Amazon and includes product review data. We use the interactions and setting following the previous work [33].

- Food[8] is also one of the series of grocery and gourmet food of Amazon that has more than 120K users and 40K items in a large-scale user-item interaction graph.
- Game[9] is an amazon-video-games review dataset that is released in 2018.
- Yelp[10] includes interaction between users and stores (e.g., restaurants, bars and so on) in business domains. We use the Yelp2018 dataset and follow the default split setting of DirectAU [33].

**Evaluation Metrics.** We use widely adopted two evaluation metrics Recall@$K$ and Normalized Discounted Cumulative Gain (NDCG@$K$) to evaluate top-$K$ recommendation performance. Here, we conduct evaluated experiments for the value of $K$ in the range of $\{10, 20, 50\}$, and report the results of $K = 20$ for simplicity. We split the interactions of each user into training/validation/test sets with a ratio of 0.8/0.1/0.1 following the previous work. In the evaluation process, we select all items except the training items for each user to calculate the evaluation metrics and report the average value of all test users' results as the final result. To validate the significant improvement of our method, we repeat our method and the sub-optimal method 5 times with different random seeds and calculate the t-test value shown in Table 4 that are less than 0.05.

We use two wide metrics to conduct top-$K$ experiments for RecDCL. The description of evaluated metrics can be described as follows:

- Recall demonstrates the ratio of recommendation items to test items. The calculation process is described as:

$$Recall@20 = \frac{|R(u)| \cap |T(u)|}{|T(u)|}, \quad (22)$$

where $|R(u)|$ and $|T(u)|$ are the recommendation and test item set for user $u$, respectively.

- NDCG indicates the importance of higher-ranked true positives. The NDCG@$K$ is calculated by Eq. 23

$$NDCG@K = \frac{DCG@K}{IDCG@K}, \quad (23)$$

where IDCG@$K$ denotes the ideal cumulative gain. The DCG@$K$ can be calculated as:

$$DCG@K = \frac{1}{|\mathcal{U}|} \sum_{u \in \mathcal{U}} \sum_{k=1}^{K} \frac{2^{(rel_{k,u})} - 1}{log_2(2 + k)}, \quad (24)$$

where $rel_{k,u}$ is 1 if $k$-th item $k$ is positive for user $u$ else it is 0.

**Implementation Details.** For fair comparison, we use the RecBole [48] framework to implement all experiments. Specifically, we initialize the parameters by Xavier initialization [10] and use the Adam optimizer [20] with a learning rate of 0.001 for all methods. The training batch size is set to 256 on Beauty and 1024 on Food, Game, and Yelp datasets. Considering the trade-off between memory cost and performance improvement, the embedding size is tuned among $\{32, 64, 128, 256, 512, 1024$ and $2048\}$. In RecDCL, the default encoder is a 2-layer LightGCN that propagates the interactions between

---

[7]https://jmcauley.ucsd.edu/data/amazon/links.html

[8]https://nijianmo.github.io/amazon/index.html
[9]https://nijianmo.github.io/amazon/index.html
[10]https://www.yelp.com/dataset

users and items. The weight $\gamma$ of $\mathcal{L}_{\text{UIBT}}$, coefficient $\alpha$ of $\mathcal{L}_{\text{UUII}}$ and the coefficient $\beta$ of $\mathcal{L}_{\text{AUG}}$ are tuned in the range of {0.005, 0.01, 0.05, 0.1}, {0.2, 0.5, 1, 2, 5, 10} and {1, 5, 10, 20}, respectively. We tuned the momentum value $\tau$ of $\mathcal{L}_{\text{AUG}}$ within {0.1, 0.3, 0.5, 0.7, 0.9}. For all baselines, the setting of specific hyper-parameters refers to their respective original work.

## C.2 Algorithm and Complexity

**Time Complexity.** We analyze how the feature-wise or batch-wise objectives impact the complexity of CL-based recommendation methods with LightGCN as the based encoder. The time complexity of RecDCL training mainly includes the following parts:

- Encoding. The time complexity of the graph convolution module in the based encoder is $O(2|\mathcal{E}|sLF\frac{|\mathcal{E}|}{B})$.

- Evaluating CL loss. For feature-wise alignment, we use only user-item positive pair, thus the complexity is linear to $\frac{|\mathcal{E}|}{B}$, i.e., $O(sF\frac{|\mathcal{E}|}{B}\frac{|\mathcal{E}|}{B})$. As defined in Eq. 6, we calculate feature-wise uniformity for users and items. The complexity of user side and item side are both $O(sF\frac{|\mathcal{E}|}{B}\frac{|\mathcal{E}|}{B})$ during whole training phase. Within a batch, the complexity of output augmentation is $O(sF)$.

Therefore, the total complexity of RecDCL is linear to the feature size $F$, i.e., $O(\frac{(2sLB+3)F|\mathcal{E}||\mathcal{E}|}{BB})$.

## C.3 Implementation Note

**Running Environment.** We implement RecDCL on PyTorch 1.9.1+cu111 and Python 3.9.7. All experiments on the Food dataset are conducted on Ubuntu with NVIDIA-A100 GPU, other experiments on the other three datasets are performed on a worker equipped with GeForce-RTX-3090.

**Projector in Figure 2.** Especially, the components of Projector in Figure 2 for feature-wise objectives are shown in Figure 3.

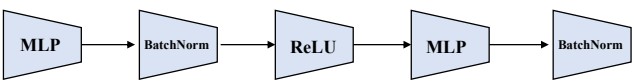

**Figure 3: The components of Projector.**

**Detailed Functions** We list the detailed implementation process of each objective function in Algorithm 2.

## C.4 Efficiency Analyses

RecDCL explores the influence of representation size from low dimension 32 to high dimension 2048 in feature-wise objectives. Therefore, we compare the training efficiency of RecDCL with the two methods that are most related to RecDCL, i.e., representative LightCCN and the state-of-the-art DirectAU. Considering space constraints and simplicity, here, we only present the largest dataset Food by Table 9. As we can see, Table 9 shows the reported results of memory consume (Memory), training time per epoch (Time/epoch), total epochs (#Epochs) and final performance (NDCG@20) when embedding size is setting as 2048. To fairly compare, we set the

---

**Algorithm 2:** The training process of the RecDCL.

**Input:** $B$: batch_size, $d$: dimension size, $e$: exponent, $\gamma$: coefficient of $\mathcal{L}_{UIBT}$, $\alpha$: coefficient of $\mathcal{L}_{UUII}$, $\beta$: coefficient of $\mathcal{L}_{AUG}$, $h$: multi-layer perceptron, $sg$: stop-gradient network, $S$: cosine similarity, $\tau$: value.

**Output:** encoder parameters $\theta$.

**Data:** $\mathbf{e}_u$: user embedding, $\mathbf{e}_i$: item embedding

```
/* Calculate feature-wise alignment loss          */
```
**Function** UIBT($\mathbf{e}_u, \mathbf{e}_i$):
$\quad C = mm(\mathbf{e}_u.T, \mathbf{e}_i).div(B)$
$\quad \mathcal{L}_{\text{align}} = diag(C).add\_(-1).pow\_(2).sum().div(d) + \gamma *$
$\quad\quad off\_diagonal(C).pow\_(2).sum().div(d)$
$\quad$**return** $\mathcal{L}_{\text{align}}$

```
/* Calculate feature-wise uniformity loss          */
```
**Function** UUII($\mathbf{e}_i$):
$\quad \mathcal{L}_{uni} = mm(\mathbf{e}_i.T, \mathbf{e}_i).add\_(c).pow\_(e).mean().log()$
$\quad$**return** $\mathcal{L}_{uni}$

```
/* Calculate batch-wise augmentation loss          */
```
**Function** BCL($\mathbf{e}_u, \mathbf{e}_i$):
$\quad \hat{\mathbf{e}}_i = \tau \mathbf{e}_i^{(l-1)} + (1-\tau)\mathbf{e}_i^{(l)}$
$\quad \mathcal{L}_{aug} = S(h(\mathbf{e}_u), sg(\hat{\mathbf{e}}_i))$
$\quad$**return** $\mathcal{L}_{aug}$

**for** *each mini-batch with positive pairs* $(u, i)$ **do**
$\quad \mathcal{L}_{\text{UIBT}} = UIBT(\mathbf{e}_u, \mathbf{e}_i)$
$\quad \mathcal{L}_{\text{UUII}} = UUII(\mathbf{e}_u)/2 + UUII(\mathbf{e}_i)/2$
$\quad \mathcal{L}_{\text{BCL}} = BCL(\mathbf{e}_u, \mathbf{e}_i)/2 + BCL(\mathbf{e}_i, \mathbf{e}_u)/2$
$\quad \mathcal{L} = \mathcal{L}_{UIBT} + \alpha * \mathcal{L}_{UUII} + \beta * \mathcal{L}_{BCL}$

---

training batch size as 1024 and implement representative methods under the same framework RecBole.

The statistic results in Table 9 show that LightGCN is the slowest in terms of the training time per epoch, which is due to the fact that it performs multi-hop neighborhood aggregation and linear propagation and optimizes objective by time-consuming BPR loss. By contrast, our RecDCL consumes the least amount of time per epoch, but more total training time due to more epochs required than LightGCN and DirectAU. Surprisingly, these two methods take roughly the same amount of space, which indicates the advantage of RecDCL over LightGCN. In fact, DirectAU is indeed the most efficient of the two indicators, i.e., training time per epoch and total epochs, but it consumes a little more memory than LightGCN and our RecDCL.

Though RecDCL requires more training epochs, it consistently brings performance improvements to RecDCL. However, possibly due to that RecDCL uses a comprehensive loss function leading to a more intricate "optimization path". In contrast, the performance of LightGCN and DirectAU plateaus quickly but keep unchanged after that. Thus, we believe the performance benefits and running costs in practice are indeed justified and acceptable.

**Table 9: Efficiency comparison of largest dataset Food (h: hour, m: minute, s: second).**

| Model | Memory | Time/epoch | #Epochs | NDCG@20 |
|---|---|---|---|---|
| LightGCN | 20,391MB | 487s | 57 | 16.77 |
| DirectAU | 26,533MB | 426s | 43 | 22.41 |
| RecDCL | 21,853MB | 318s | 253 | 23.27 |

## C.5 Study of RecDCL on Food and Game

Similarly, to verify the effectiveness of each component, we show the statistical results of the ablation study on two datasets, Food and Game, as presented in Table 10. From this table, we can find that removing either or both of the three components in RecGCL leads to decreasing performance, while the five variants (except "w/ UUII") can do better than the baseline LightGCN in general. It demonstrates that the dual CL design of RecDCL will benefit the performance in graph collaborative filtering. Besides, feature-wise CL and batch-wise CL complement each other and improve the performance in different aspects.

**Table 10: Performance comparison of different designs of RecDCL on Food and Game.**

| Method | Food | | Game | |
|---|---|---|---|---|
| | R@20 | N@20 | R@20 | N@20 |
| LightGCN | 24.56 | 16.77 | 19.20 | 8.91 |
| w/ UIBT | 23.30 | 16.70 | 18.12 | 8.62 |
| w/ UUII | 4.84 | 3.15 | 1.96 | 0.63 |
| w/ BCL | 28.01 | 23.01 | 20.25 | 9.85 |
| w/ UIBT & UUII | 26.57 | 20.55 | 18.18 | 8.80 |
| w/ UIBT & BCL | 28.69 | 23.12 | 20.39 | 9.85 |
| w/ UUII & BCL | 26.53 | 20.51 | 19.32 | 9.21 |
| RecDCL | **28.95** | **23.27** | **20.44** | **9.87** |
| %Improv. | 17.87% | 38.76% | 6.46% | 10.77% |

## C.6 Hyper-parameter Sensitivity of RecDCL on Beauty and Yelp

**Effect of embedding size $F$.** Generally, the embedding size is set as 64 by default in most CF methods [14, 33]. To validate the impact of feature-wise objectives, we run all methods with different embedding sizes from 32 to 2048 on four datasets and show the best results of each dimension on Beauty and Yelp in Figure 4. We have the following observations:

(1) as embedding size increases, RecVAE and our RecDCL continue to grow, while the performance of other models in terms of Recall@20 first improves, and then holds the line even degrades, even though we provided these methods with exactly the same experimental settings. For larger F ( > 2048), the results of RecDCL still increase, while LightGCN degrades the performance and DirectAU causes the out-of-memory problem. In this sense, instead of an

unsuitable comparison, the above phenomenon may also be understood as a property of RecDCL, i.e., it can benefit from increasing embedding size but the baseline methods failed. In fact, the above setting has been used in literature. For example, in [44], although a promising self-supervised method BarlowTwins only surpasses the baselines at a large embedding size, it has received extensive attention and inspired many follow-up works.

(2) Though RecVAE has the same trend, RecDCL exceeds RecVAE on these two datasets and has an improvement of up to 23.67% on Beauty. This again demonstrates that RecDCL can capture the information of high-dimensional representation well.

(3) Indeed, LightGCN enjoying the joint advantages of high-order neighborhood information and low-rank MF should be expected to achieve better performance than BPR-MF, which is actually consistent with our experimental observation when the embedding size F is not large (e.g., 32 and 64 on Beauty). Furthermore, these experimental results for small embedding sizes are also in agreement with that reported in the literature [33]. However, since this paper focuses more on the FCL objective whose effectiveness in empowering CF necessitates a larger embedding size, we regard the embedding dimension as a tunable hyperparameter. In this situation, LightGCN does not benefit much from the increase in embedding size despite our best efforts in parameter tuning. As a result, it lags behind BPR-MF at large embedding sizes on two (out of four) datasets (Beauty and Food). On the other hand, by searching throughout the hyperparameter space of embedding size, NGCF showcases slight advantages over LightGCN on the Food dataset. However, on the other three benchmarks, the performance of NGCF is only comparable to or inferior to that of LightGCN.

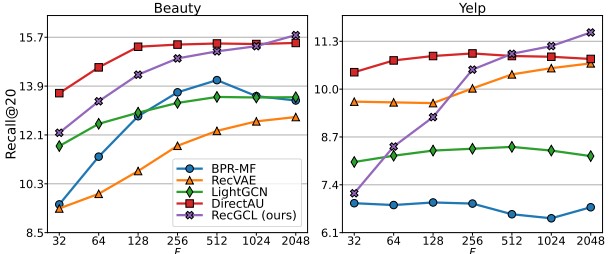

**Figure 4: Recall@20 results of different embedding sizes of representative baselines and RecDCL on Beauty and Yelp.**

**Effect of coefficient $\alpha$ of UUII.** Similarly, we analyze the impact of coefficient $\alpha$ in the range of {0.2, 0.5, 1, 2} of UUIInomial within users and items on the Beauty and Yelp dataset in Figure 5. (a) and in Figure 5. (b), respectively. Obviously, the small value, i.e., 0.2, will promote the model performance on the Beauty dataset, but the trend is just the opposite on Yelp. Obviously, the best value of $\alpha$ is 0.2 on Beauty and 2 on Yelp, which indicates that different datasets have different optimal situations.

**Effect of coefficient $\beta$ of BCL.** As verified in Section 4, batch-wise output augmentation plays a critical role in SSL-based recommendation. Consequently, it is necessary for us to show the model performance $w.r.t.$ $\beta$ in Figure 6. According to this figure, we can observe that: (1) after adding batch-wise augmentation, the whole

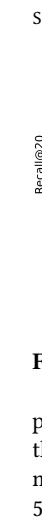

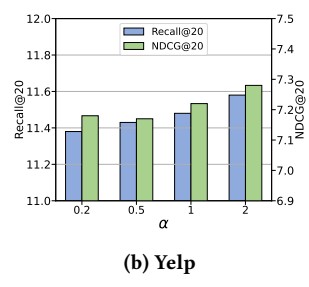

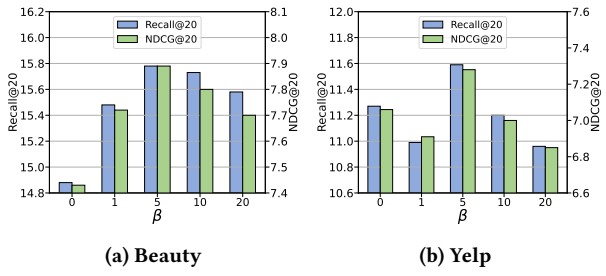

**Figure 5: Influence of different $\alpha$ of UUII on Beauty and Yelp.**

performance is better than only "with UIBT & UUII". (2) Increasing the value of $\beta$, e.g., 20, will keep consistent or lead to poor performance. (3) $\beta$ is consistent on different datasets, which is setting as 5 on Beauty and Yelp.

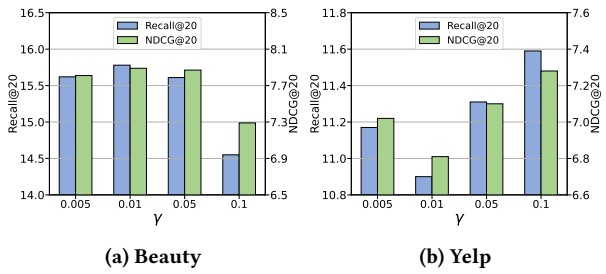

**Figure 6: Influence of different $\beta$ of BCL on Beauty and Yelp.**

**Effect of coefficient $\gamma$ of UIBT.** We explore the sensitivity of RecDCL to the coefficient $\gamma$ of UIBT, which trades off the desiderata of invariance term and informativeness of the representations. Figure 7 shows that the influence of UIBT's coefficient $\gamma$ in range of {0.005, 0.01, 0.05, 0.1} in terms of Recall@20 and NDCG@20 on Beauty and Yelp. Obviously, We can find that our RecDCL is not very sensitive to this hyperparameter. Besides, UIBT's coefficient varies on different datasets. More specially, the optimal coefficient is 0.01 on Beauty dataset in Figure 7. (a) while the best $\gamma$ is 0.1 on the Yelp dataset in Figure 7. (b).

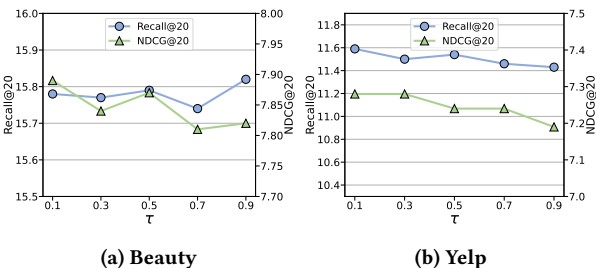

**Figure 7: Influence of different $\gamma$ of UIBT on Beauty and Yelp datasets.**

**Effect of $\tau$ of BCL.** We study the effect of $\tau$ in rang of {0.1, 0.3, 0.5, 0.7, 0.9} of batch-wise output augmentation and report the results in Figure 8. We find that: (1) $\tau$ is sensitive to Beauty. For example,

the $\tau$ set as 0.7 performs better than other values. (2) In contrast, fixing $\tau$ to three values will rarely affect performance on the Yelp dataset. In a nutshell, we suggest tuning $\tau$ in the range of 0.1 ~ 1.0 carefully.

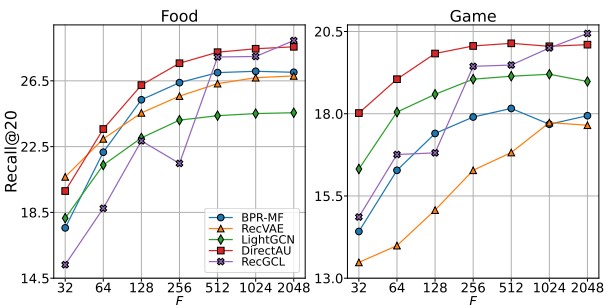

**Figure 8: Influence of different $\tau$ of BCL on Beauty and Yelp.**

## C.7 Hyper-parameter Sensitivity of RecDCL on Food and Game

**Effect of embedding size $F$.** Considering that feature-wise objectives are a key point of our method, embedding size $F$ is a critical hyper-parameter that affects the performance. Thus, it is essential to show the changing trends of embedding size $F$ from 32 to 2048 on the remaining two datasets Food and Game in Figure 9. As we expected, the performance of almost all methods increases with $F$, which also verifies that the larger $F$ value can indeed provide richer representation information and feature-wise CL method can achieve ideal performance while embedding size is high-dimensional. Among them, the most significant improvement is our RecDCL. More specifically, when $F$ is 32, our method lags behind other methods (especially on Food), but our method is the best when d is 2048, improving by 110.34% on Food and 36.67% on Game.

**Figure 9: The experimental results of different embedding sizes of baselines and our RecDCL on Food and Game.**

**Effect of coefficient $\gamma$ of UIBT.** As described in Figure 10, we test the effect of $\gamma$ in UIBT on the performance of RecDCL on Food and Game in terms of Recall@20 and NDCG@20, in which $\gamma$ is in the range of {0.005, 0.01, 0.05, 0.1} like the other two datasets Beauty and Yelp. Although Recall@20 and NDCG@20 on both datasets Food and Game basically show a trend of rising first and then falling,

the magnitude of the change is not particularly obvious, which indicates that it is not very sensitive to the parameter $\gamma$. Moreover, we can find optimal $\gamma$ for different datasets. Specifically, the optimal $\gamma$ is 0.05 on Food and 0.01 on Game, respectively.

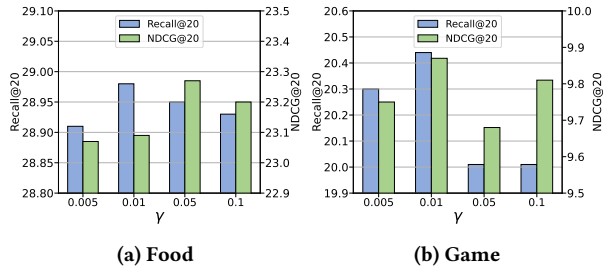

**(a) Food**     **(b) Game**

**Figure 10: Influence of different $\gamma$ of UIBT on Food and Game datasets.**

**Effect of coefficient $\alpha$ of UUII.** To more comprehensively investigate the influence of $\alpha$ on the effectiveness of RecDCL, we further vary $\alpha$ in {0.2, 0.5, 1, 2} on Food and Game two datasets, and statistical results are displayed in Figure 11. (a) and in Figure 11. (b), respectively. Obviously, we can see that the performance of the two datasets shows opposite trends. Specifically, increasing $\alpha$ generally improves the performance on Food while leading to poor performance on Game. Therefore, the optimal performance on Food is obtained when $\alpha$ is 1, while the best value of $\alpha$ on Game is 0.2.

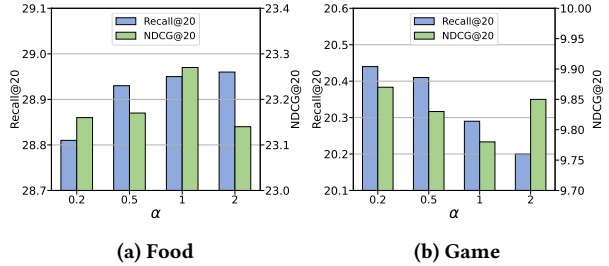

**(a) Food**     **(b) Game**

**Figure 11: Influence of different $\alpha$ of UUII on Food and Game.**

**Effect of $\tau$ of BCL.** Similarly, we also consider varying $\tau$ in the range of 0.1 to 0.9 in steps of 0.2 and then illustrate the changing curves of Recall@20 and NDCG@20 on Food and Game in Figure 12. From Figure 12.(a) and Figure 12.(b), we can observe that: (1) Compared with hyperparameters $\gamma$ and $\alpha$, $\tau$ of BCL is relatively sensitive on two datasets. Thus, we believe that it is necessary for us to carefully tune $\tau$ in the range of $0.1 \sim 0.9$ carefully. (2) The changing trends in the two datasets are basically the same. Especially, setting $\tau$ to 0.1, 0.5, 0.7, or 0.9 will rarely affect the performance of RecDCL. Besides, the $\tau$ is 0.3 and performs better than other values.

**Effect of coefficient $\beta$ of BCL.** As analyzed in the previous body text, batch-wise output augmentation is essential for SSL-based recommendation. Thus, it is also necessary to test the effect of coefficient $\beta$ on the performance of RecDCL. Illustratively, we present the results $w.r.t.$ different $\beta$ in the range of 0 to 20 in

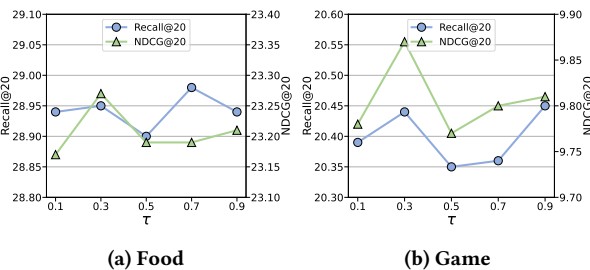

**(a) Food**     **(b) Game**

**Figure 12: Influence of different $\tau$ of BCL on Food and Game.**

variable steps in Figure 13. According to Figure 13a and Figure 13b, we can observe that: (1) Compared with only "with UIBT & UUII" (i.e., $\beta = 0$), batch-wise augmentation (i.e., $\beta > l0$) improve the whole performance by 8.64%, 9.43% on Food and 12.64%, 17.02% on Game in terms of Recall@20 and NDCG@20, respectively. (2) Unlike Beauty and Yelp two datasets, larger values first lead to better performance on Food and Game and then decrease in general. (3) The impact of $\beta$ on RecDCL is consistent on different datasets, which is both set as 10 on Food and Game datasets.

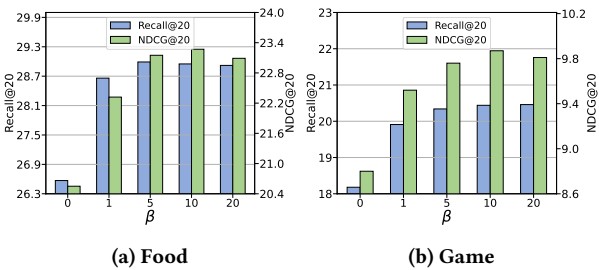

**(a) Food**     **(b) Game**

**Figure 13: Influence of different $\beta$ of BCL on Food and Game.**

## C.8 Detailed Hyper-parameter on Four Datasets

We list the respective hyper-parameter on each dataset with the best performance in Table 11.

**Table 11: Best hyperparameter setting.**

| Dataset | Emb size | $\gamma$ | $\alpha$ | $\tau$ | $\beta$ |
|---------|----------|----------|----------|--------|---------|
| Beauty  | 2048     | 0.01     | 0.2      | 0.1    | 5       |
| Food    | 2048     | 0.05     | 1        | 0.3    | 10      |
| Game    | 2048     | 0.01     | 0.2      | 0.3    | 10      |
| Yelp    | 2048     | 0.1      | 2        | 0.5    | 1       |

