# OpenReview forum: "RecDCL: Dual Contrastive Learning for Recommendation"
_ACM.org/TheWebConf/2024/Conference — TheWebConf24 Oral_

### Official Review · Reviewer_qBab · 2023-11-17

**Novelty:** 5
**Technical Quality:** 5

**Review:**

This paper integrates DirectAU and Barlow Twins for the first time in the field of recommender systems, achieving outstanding results by combining BCL and FCL. It also provides some explanations. Overall, the idea presented in the paper is of interest to the community, and the amount of work done is substantial.

Some concerns are as follows.
1. The motivation of the proposed combination needs further clarification. For example, why using FCL would benefit recommendation?

2. Unlike the previous graph collaborative filtering, embedding sizes ranging from 32 to 2048 are used in experiments in Table 4. It would be better if the relationship between embedding size and performance could be given.

3. More competitors in the SSL category could be evaluated and compared.

Some minor issues:
- In section 3.3, the authors compare the average entropy of FCL, BCL and BCL+FCL embeddings on the Yelp dataset in two ways. What is the rationale behind?
- In section 4.2, the authors mention "we conduct data augmentation on output representation and generate contrastive but related views for representation learning, namely BCL." This sentence needs further explanation.
- Is the AUG in Algorithm 1 a typo?

**Questions:**

- Why not compare with more SSL baselines?

**Reviewer Confidence:**

3: The reviewer is confident but not certain that the evaluation is correct

**Scope:**

3: The work is somewhat relevant to the Web and to the track, and is of narrow interest to a sub-community

---

### Official Review · Reviewer_AgtS · 2023-11-21

**Novelty:** 5
**Technical Quality:** 7

**Review:**

This paper focuses on Contrastive Learning for recommendation. The authors propose a dual contrastive learning method for recommendation systems, referred to as RecDCL. It combines batch-wise contrastive learning (BCL for short) and feature-wise contrastive learning (FCL) to address the limitations of existing methods.

In general, the paper appears to be of high quality.
In terms of clarity, it is well-written and provides a clear explanation of the motivation, methodology, and experiments. The figures and examples help to illustrate the concepts and make the content more understandable.
Additionally, the experiments conducted in the paper are comprehensive, and they involve extensive experiments on multiple public datasets and one industrial dataset to validate the effectiveness of the proposed method.
The main advantages are as follows:
1. The paper provides a clear explanation of the proposed method, RecDCL, and its benefits.
2. The ablation experiment in the paper proves the better performance due to the design of the model structure.
3. The motivation of the paper can be well reflected in the methods and experiments.
4. The paper provides a theoretical analysis of the connection between feature-wise and batch-wise objectives, contributing to the understanding of contrastive learning in recommendation.

**Questions:**

Can you elaborate on the theoretical foundation of employing average entropy to assess the embedding probability distributions? Specifically, how does this method effectively evaluate the characteristics of the embedding space created by BCL, FCL, and their combination?

In Equation 7, a more detailed explanation of the DirectAU component and its specific contribution to the loss function would be valuable(since it comes from another work[Towards Representation Alignment and Uniformity in Collaborative Filtering]). Briefly clarifying the role and theoretical basis of DirectAU would enhance the overall understanding of your methodology.

**Reviewer Confidence:**

4: The reviewer is certain that the evaluation is correct and very familiar with the relevant literature

**Scope:**

4: The work is relevant to the Web and to the track, and is of broad interest to the community

---

### Official Review · Reviewer_nF3F · 2023-11-22

**Novelty:** 4
**Technical Quality:** 5

**Review:**

This paper discusses the effect of the combination of batch-wise contrastive learning (I prefer to call it instance-wise contrastive learning, which could be more accurate) and feature-wise contrastive learning (it's farfetched to consider Barlow Twins a CL-based method) on improving recommendation. In many previous studies, the effectiveness of BCL based on in-batch negative sampling has been demonstrated, but FCL is relatively new to self-supervised recommendation. This paper also provides theoretical analysis to connect BCL and FCL, which is commendable. The experiments look good and have verifed their ideas. However, I have the following concerns on this paper.

 + First, the terminology used in this paper is problematic. I think it's farfetched to consider Barlow Twins a CL-based method. In the original paper of this method, Barlow Twins is only called self-supervised, neither do BYOL/SimSiam. Similarly, the compared recommendation methods BUIR and DirectAU (even not self-supervised) in this paper are not contrastive methods either. So, I don't consider the so called FCL in this paper a contrastive loss.
 + The organization and writing of this paper should be improved. It should be self-contained, with some important parts like the theoretical analysis in the first 8 pages. The introduction is not readily understandable for new researchers working on this area and is rough. For instance, for the batch-wise contrastive learning, the terms on-diagonal and off-diagonal are too technical. When it comes to the fomula and code implementations, it is more unstandable to use these terms. Section 3.3 is not very clear.
+ The proposed method is mainly based SimSiam and Barlow Twins. As these two methods are not considered contrastive, the proposed method is only self-supervised instead of constrastive. The technical contribution is somehow limited due to the terminology error. I expected more than a combination of two losses.
+ This paper does not compare the proposed method with the real contrastive recommendation methods like SGL and SimGCL, which are the state-of-the-art methods. According to my experience, the peformance of BUIR is incompetitive and even cannot outperform LightGCN. I wonder why it seems competitive in the experiments of this paper. It looks like the methods were not fine-tuned.

Overall, this paper has some certain merits. My main concern about the paper is terminology issue. Beyond contrastive learning, this paper is still informative. Some details should be more clear and the sota methods should compared in the revised paper.

**Questions:**

+ If the authors insist that the proposed method is contrastive, can you justify your claims?
 + Can you make section 3.3 more understandable by providing clear experiment details? I can get the key points of this section but still have confusions on this.
 + Can you compare the proposed method with SGL and SimGCL?

**Reviewer Confidence:**

4: The reviewer is certain that the evaluation is correct and very familiar with the relevant literature

**Scope:**

4: The work is relevant to the Web and to the track, and is of broad interest to the community

---

### Official Review · Reviewer_64Zo · 2023-11-23

**Novelty:** 6
**Technical Quality:** 6

**Review:**

This paper investigates the combinatorial use of batch contrastive learning (BCL) and feature contrastive learning (FCL) for recommender systems. This paper starts with a toy example in 2D space showcasing why intuitively combining BCL with FCL may lead to higher quality embeddings, followed by some theoretical justifications and further experimental studies. The experiments show a pretty good coverage and are relatively comprehensive in proving the effectiveness of the proposed approach.

The highlights of this paper are:
1. The authors have provided theoretical and experimental rationale for the combinatorial use of BCL and FCL for recommendation, where an improved version of this combined CL paradigm is proposed. The approach is overall well motivated.
2. The coverage of the experiments are mostly good, where analysis on key hyperparameters across all datasets is helpful.
3. The inclusion of an industrial dataset is a nice touch.

In the meantime, I believe this paper can still be improved from the following aspects:
1. Some more technical details can be included/reorganized to make this paper more self-contained. For example, the calculation of the correlation matrix $\textbf{C}$ for FCL can be included in Section 2.3 (can be done by moving Eq.4 to this section).
2. Some design choices can be better justified. For example, in the BCL part, among many different choices like graph augmentation-based (e.g., [37]) or sampling-free BCL (e.g., [42]), it is unclear why the authors specifically use the BYOL design.
3. The selection of SSL-based baselines can be enriched by including more recent, strong baselines, such as the following:
- LightGCL: Simple Yet Effective Graph Contrastive Learning for Recommendation, ICLR'23
- An MLP-based algorithm for efficient contrastive graph recommendations, SIGIR'22
- Candidate-aware Graph Contrastive Learning for Recommendation, SIGIR'23
4. The writing and presentation of this paper can benefit from a major improvements. Some edits:
- Line 276, "be closed 0" --> close to
- Line 496-497, incomplete sentence
- Line 570, "embedding generate by" --> generated
- There are some inconsistent notations, e.g., Eq.5 uses non-bold $C$ but bold $\textbf{C}$ is used elsewhere.

**Questions:**

1. How sensitive the FCL component is to the batch size B?

2. What is the advantage of the parameter bootstrapping for the BCL in the proposed solution, compared with other BCL methods?

**Reviewer Confidence:**

4: The reviewer is certain that the evaluation is correct and very familiar with the relevant literature

**Scope:**

4: The work is relevant to the Web and to the track, and is of broad interest to the community

---

### Official Review · Reviewer_VU3t · 2023-11-25

**Novelty:** 6
**Technical Quality:** 7

**Review:**

This paper introduces a dual contrastive learning method called RecDCL that is motivated through analysis of what happens in Batch-wise CL and Feature-wise CL that eliminates redundancy both between and within users and items in a recommendation setting. The work is extremely well motivated through deep analyses of previous works, and is easy to follow.

The work builds on established methods and provides good intuition. The authors design an FCL objective and a BCL objective that capture embedding importance as well as provide all the theoretical benefits they mention in their analysis from a dual CL.

The authors also do a very good job of anonymizing all the code while providing all the source code and analyses of this work. The experiments seem thorough but there could be more experiments done with SSL baselines.

**Questions:**

Something that wasn't clear to me is how the trade off hyperparameters \alpha and \beta would impact the work. Does it control how close we want to get to true orthogonality? What are some of the trade-offs that people should have in mind for this? How would the batch size impact this?

**Reviewer Confidence:**

3: The reviewer is confident but not certain that the evaluation is correct

**Scope:**

4: The work is relevant to the Web and to the track, and is of broad interest to the community

---

### Decision · Program_Chairs · 2024-01-22

**Decision:**

Accept (Oral)

**Comment:**

The paper investigates the synergistic application of Batch Contrastive Learning (BCL) and Feature Contrastive Learning (FCL) in the realm of recommender systems. The research problem tackled in this study is notably novel, and the motivation and rationale are well justified through a combination of intuitive examples and a thorough theoretical and experimental analysis. The authors have undertaken extensive experiments to validate their proposed approach, and the results are promising, providing substantial support for the claims put forth in the paper.

  The overall presentation of the paper could benefit from significant improvement. There is ample room for enhancement, particularly in terms of incorporating additional technical details or reorganizing existing information to bolster the self-contained nature of the manuscript. Furthermore, while the design choices made are outlined, a more comprehensive justification of these decisions would contribute to a clearer understanding for the reader. Providing a deeper insight into the rationale behind the chosen methodologies and strategies will enhance the overall coherence of the paper. In terms of baselines, the suggestion to enrich the selection of Semi-Supervised Learning (SSL)-based counterparts with more recent and robust alternatives is well-founded. This addition would not only strengthen the comparative analysis but also ensure the relevance and competitiveness of the proposed method in the current research landscape. Lastly, it is strongly recommended that the authors adhere to established conventions regarding terminology. Ensuring alignment with established terminology conventions will enhance the credibility and impact of the paper.